# Beyond Cropping and Rotation: Automated Evolution of Powerful Task-Specific Augmentations with Generative Models

## Abstract

Data augmentation has long been a cornerstone for reducing overfitting in vision models, with methods like AutoAugment automating the design of task-specific augmentations. Recent advances in generative models, such as conditional diffusion and few-shot NeRFs, offer a new paradigm for data augmentation by synthesizing data with significantly greater diversity and realism. However, unlike traditional augmentations like cropping or rotation, these methods introduce substantial changes that enhance robustness but also risk degrading performance if the augmentations are poorly matched to the task. In this work, we present EvoAug, an automated augmentation learning pipeline, which leverages these generative models alongside an efficient evolutionary algorithm to learn optimal task-specific augmentations. Our pipeline introduces a novel approach to image augmentation that learns stochastic augmentation trees that hierarchically compose augmentations, enabling more structured and adaptive transformations. We demonstrate strong performance across fine-grained classification and few-shot learning tasks. Notably, our pipeline discovers augmentations that align with domain knowledge, even in low-data settings. These results highlight the potential of learned generative augmentations, unlocking new possibilities for robust model training.

## 1 Introduction

Generative AI has rapidly advanced across multiple domains. In computer vision, diffusion models now surpass GANs in producing realistic images and videos from simple prompts (Dhariwal & Nichol, 2021). In language, models like GPT generate human-like text and code, achieving high scores on standardized tests (OpenAI et al., 2024). Similar breakthroughs extend to generative audio (Schneider, 2023) and 2D-to-3D shape generation (Karnewar et al., 2023). These advances raise an important question: to what extent can AI-generated content improve AI itself (Yang et al., 2023b)? While far from true self-improvement, generative models are increasingly influencing their own training processes.

A key challenge in leveraging synthetic data is the syn-to-real gap—the discrepancy between generated and real-world data. Poorly matched synthetic augmentations degrade performance rather than enhance it. For example, diffusion models still struggle with fine details such as realistic fingers (Narasimhaswamy et al., 2024). Thus, a model trained on data augmented by flawed synthetic images may reinforce errors. Similarly, a language model could amplify its own biases by training on text that it generated itself. This issue is particularly critical in tasks requiring fine-grained distinctions, such as image classification, or in low-data settings like few-shot learning. Addressing this gap is essential for generative augmentations to contribute meaningfully to AI training.

Hence, methods that use synthetic or simulated data must balance the tradeoff between data variability and fidelity. This can be achieved by constraining data generation to closely match the real-world distribution, thereby reducing its variability while improving its fidelity. This approach has been successful in fields like robotics (Lu et al., 2024) and autonomous vehicles (Song et al., 2024). However, it has only seen limited application in synthetic image generation for computer vision. This work tackles the challenge of fine-grained few-shot classification. Due to the lack of real samples, syn-

thetic data provides an attractive option for boosting performance. Since fine-grained distinctions between classes can be easily missed, a carefully designed image generation pipeline is required.

We propose using generative AI not for data creation, but for data augmentation—a paradigm shift. Instead of generating data from scratch, we condition the process on real data, thereby ensuring that it preserves the semantic priors and underlying structure of the original distribution while introducing meaningful and novel variations. While this approach constrains synthetic data to resemble real data, it also provides stronger guarantees of its validity, effectively overcoming the syn-to-real gap.

Motivated by this vision, we design EvoAug, a pipeline that automatically learns a powerful augmentation strategy. Our work makes use of evolutionary algorithms, which have been shown to work in a variety of domains and still remain more sample-efficient and straightforward than other methods (Ho et al., 2019; Wang et al., 2023). This is especially important when dealing with complex augmentation operators like conditional diffusion and NeRF models, where evaluation is expensive, gradients are very difficult to approximate, and sample efficiency is paramount.

As part of our pipeline, we construct an augmentation tree—a binary tree that applies a series of augmentation operators in accordance with learned branching probabilities. The augmentation tree can then be used to produce synthetic or augmented variations of the images in the dataset by stochastically following root-to-leaf paths. Our trees include nodes that perform either classical or generative augmentations. To produce accurate synthetic data, we condition the diffusion models on existing structural and appearance-based information rather than solely relying on prompt-based image generation. Our approach is powerful enough to work even with very small datasets and provides promising results on fine-grained and few-shot classification tasks across multiple datasets.

Our main contributions are the following:

1. The first automated augmentation strategy to leverage both modern augmentation operators like controlled diffusion and NeRFs, along with traditional augmentation operators like cropping and rotation

2. Strong results on fine-grained few-shot learning, a challenging domain where prior work has failed to preserve the minor semantic details that distinguish the classes

3. Novel unsupervised strategies that scale as low as the one-shot setting, where no supervision to evaluate augmentations is available

4. Constructing an augmentation pipeline from only open-source, pre-trained diffusion models, without requiring domain-specific fine-tuning

## 2 RELATED WORK

Data augmentation reduces model overfitting by applying image transformations that preserve the original semantics while introducing controlled diversity into the training set. Traditional augmentations include rotations, random cropping, mirroring, scaling, and other basic transformations. These straightforward techniques remain fundamental in state-of-the-art image augmentation pipelines. More advanced methods—such as erasing (Zhong et al., 2020; Chen et al., 2020; Li et al., 2020; DeVries & Taylor, 2017), copy-pasting (Ghiasi et al., 2021), image mixing (Zhang et al., 2017; Yun et al., 2019), and data-driven augmentations like AutoAugment (Cubuk et al., 2018) and its simplified variant RandAugment (Cubuk et al., 2020)—have expanded the augmentation toolbox.

Another approach involves generating synthetic data using generative models (Figueira & Vaz, 2022). Early work explored GANs (Besnier et al., 2020; Jahanian et al., 2021; Brock et al., 2018), VAEs (Razavi et al., 2019), and CLIP (Ramesh et al., 2022), achieving strong results (Engelsma et al., 2022; Skandarani et al., 2023). Recently, diffusion models, particularly for text-to-image synthesis, have surpassed GANs in producing photorealistic images (Nichol et al., 2021; Ramesh et al., 2022; Saharia et al., 2022b; Yang et al., 2025). Trained on large-scale internet data (Schuhmann et al., 2022), diffusion models have been used for augmentation (Azizi et al., 2023; Sarıyıldız et al., 2023; He et al., 2022; Shipard et al., 2023; Rombach et al., 2022; Islam et al., 2025; 2024), often relying on class names or simple class agnostics prompts to guide generation. Despite promising initial results, synthetic data remains inferior to real data, highlighting the persistent domain gap between the two (Yamaguchi & Fukuda, 2023).

To address this gap, recent approaches have incorporated conditioning the generative process on real data. Some popular methods involve projecting the original images to the diffusion latent space (Zhou et al., 2023), fine-tuning diffusion models on real data (Azizi et al., 2023), leveraging multi-modal LLMs to obtain detailed, custom image captions for high-quality text prompting(Yu et al., 2023), and employing image-to-image diffusion models that enable direct conditioning on a specific image (Saharia et al., 2022a; Meng et al., 2021; Zhang et al., 2023; He et al., 2022; Trabucco et al., 2025). Controlled diffusion, a subset of these methods, introduces a more powerful paradigm, furthering the efficient use of both text and image priors (Fang et al., 2024; Islam & Akhtar, 2025) with applications in segmentation (Trabucco et al., 2023) and classification (Goldfeder et al., 2024) problems.

Given such a wide range of augmentation operators, an important problem is knowing which augmentations to use for a specific task, without the use of domain knowledge. This task, of automatically learning augmentation policies, falls under the class of meta learning and bi-level optimization problems, where we seek to learn a component of the learning algorithm itself (Hospedales et al., 2021). These algorithms generally fall under one of the following categories: gradient-based optimization, RL-based optimization, Bayesian optimization, and evolution-based optimization.

In the context of learning augmentation policies, all these methods have seen success (Yang et al., 2023a). Differentiable methods often train a neural network to produce augmentations (Lemley et al., 2017), sometimes in a generative adversarial setup (Shrivastava et al., 2017; Tran et al., 2017). By far the most notable method, AutoAugment (Cubuk et al., 2018), employs reinforcement learning. While RL is traditionally sample inefficient, improvements upon vanilla RL strategies have leveraged Bayesian methods (Lim et al., 2019), evolutionary strategies (Ho et al., 2019; Wang et al., 2023), or approximate gradient estimation for first-order optimization (Hataya et al., 2020).

Learning augmentation policies is especially challenging in low data settings, as full data policies are usually not transferable to the few-shot case. Various approaches have been considered, including proposing K-fold validation as a method of retaining the data while still performing validation (Naghizadeh et al., 2021). However, this method does not scale to one-shot settings. Utilizing clustering as a label-efficient evaluation method, where augmentations are designed to stay within their corresponding class cluster, can address this limitation (Abavisani et al., 2020).

## 3 METHODS

### 3.1 AUGMENTATION OPERATORS

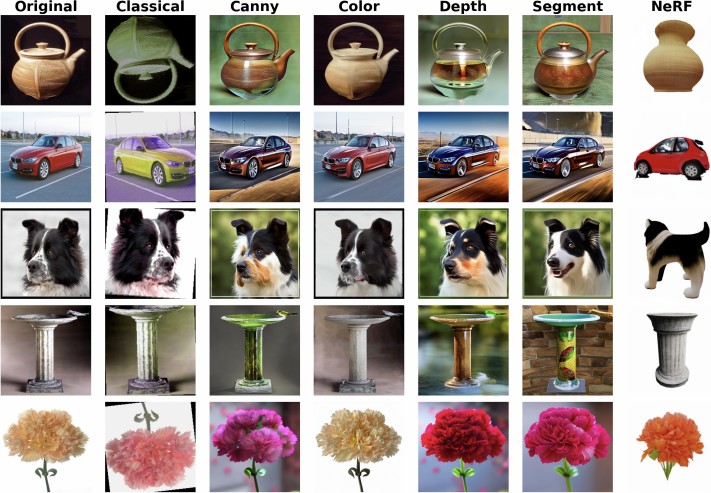

Figure 1: Example image augmentations using our pipeline. Classical augmentations include color jitter, rotation, and random cropping. Canny, color, depth, and segment use existing image information to steer a ControlNet diffusion model. NeRF uses a zero-shot NeRF to perform a 3D rotation.

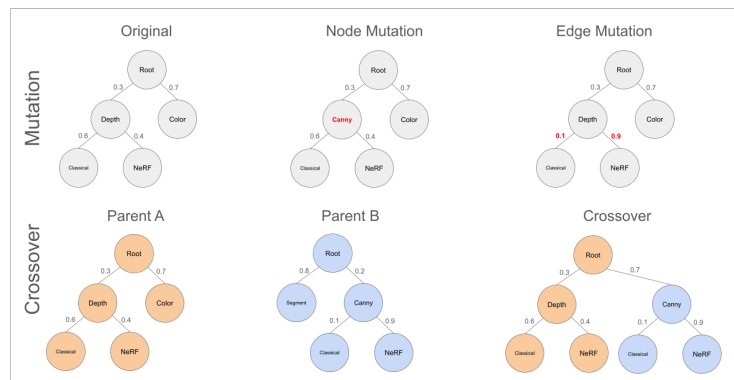

Figure 2: Mutation and Crossover for Augmentation Trees

The generative augmentation operators are based on both diffusion and NeRFs. For diffusion-based operators, we use ControlNet (Zhang et al., 2023), an architecture which allows rapid customization of diffusion models without fine-tuning. To condition the model, we extract edges using Canny edge detection (Canny, 1986), segmentations using Segment Anything (Kirillov et al., 2023), depth maps using MiDaS (Ranftl et al., 2020), and color palettes by simply downsampling the image. This gives four diffusion-based augmentation operators, termed "Canny", "Segment", "Depth" and "Color". We use Zero123 (Liu et al., 2023b) for NeRF-based augmentation. This model creates a 3D reconstruction of an image from a single shot, allowing for 3D rotation. We then rotate 15 degrees left or right when performing an augmentation using this model. We term this operator "NeRF". Next, we include another augmentation operator, termed "Classical." This includes the full set of traditional augmentations: random crop, translation, scale, rotation, color jitter, and flip. This operator allows the evolution process to decide whether to include and build on the traditional classical augmentation pipeline or exclude it. Sometimes, all augmentations can be harmful, so we also included a "NoOp" operator that simply duplicates the existing image. Figure 1 gives examples of these operators.

## 3.2 EVOLUTIONARY STRATEGY

For our augmentation policy learning pipeline, we choose an evolutionary approach. This choice is motivated by practical considerations: diffusion and NeRF based augmentation is considerably more expensive to evaluate than traditional augmentations, so pipeline efficiency is crucial. Population-based evolutionary strategies have been shown to be as effective as RL approaches, with less than one percent of the computational effort (Ho et al., 2019). While gradient approximation methods have been shown to be even more efficient in some cases(Hataya et al., 2020), those results are for approximating gradients of simpler transformations, and do not translate to our pipeline, which can handle arbitrary generative modules. Further, recent work has shown evolution to be effective for searching for augmentation polices even in very complex augmentation spaces (Wang et al., 2023).

We define an augmentation tree as a binary tree, where each node represents an augmentation operator. The edges of our tree represent transition probabilities to each child node, summing to 1. This structure is chosen as it serves as a common genome for evolutionary algorithms.

**Mutation** Illustrated in figure 2, mutation can occur at either the node level or the edge level. An edge mutation reassigns the probabilities of a transition between two child nodes. A node mutation switches the augmentation operator of that node (eg. Depth node becomes a Canny node).

**Crossover** Also illustrated in figure 2, crossover is the other basic evolutionary operator. Two parents are selected, a child is created by splicing the branches of the parents together.

We thus define a population $P$ of size $n$, of initial trees. In each generation, we use mutation and crossover to generate $c$ children $P_{new}$, that are appended to $P$. Finally, the population is evaluated with a fitness function $f$, and the top $n$ are kept for the next generation. Mutation and crossover probability are parameterized by $p_m$ and $p_c$ respectively. Algorithm 1 describes this process.

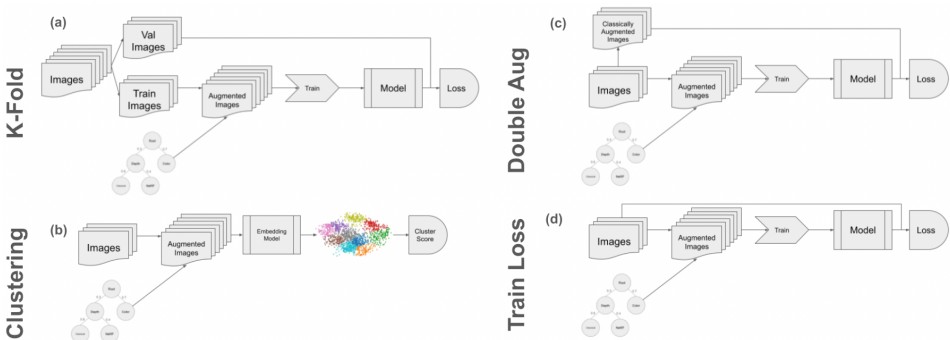

Figure 3: Tree Learning Pipelines. (a) K-Fold applies when there is more than one example per class. (b) We can measure cluster quality for the 1-shot case. (c) We can duplicate the image and assume the problem to be 2-shot instead of 1-shot (d) We can simply use training loss, though it is risky to assume that lower train loss equates to better performance.

### 3.3 FITNESS FUNCTIONS

The goal of our augmentation strategy is to improve downstream model robustness, and thus the fitness function we choose to evaluate augmentation trees should either directly reflect what we seek to achieve or be a strong proxy. Note that in a full data setting, training data can be split into a train and validation. An augmentation tree can be evaluated by simply training a model with generated augmentations on the training data and measuring performance on the previously unseen evaluation data. We divide our discussion into two, more difficult, settings.

#### 3.3.1 LOW DATA SETTING

In the low-data and few-shot case, the challenge becomes managing the noise of the evaluation function. We can no longer rely on a single train/val split to accurately measure the performance of a tree as low-data settings introduce high variability in splits. Thus, we use K-fold cross-validation.

In addition, directly using accuracy as our metric is no longer appropriate, as our validation set remains small enough that accuracy becomes coarse-grained and unstable. As a result, to align with the convention of higher fitness values corresponding to better candidates in the population, we use the negative validation loss as the fitness function in these settings. Algorithm 2 describes this process. The pipeline can be seen in figure 3a.

---

**Algorithm 1** Evolutionary Search for Augmentation Trees

---

**Require:** Population size $p$, number of generations $g$, fitness function $f$, number of children $c$, mutation probability $p_m$, crossover probability $p_c$

1: $P \leftarrow$ InitializePopulation($p$)
2: **for** $i = 1$ to $g$ **do**
3: $\quad P_{\text{new}} \leftarrow$ MutateAndCrossover($P, c, p_m, p_c$)
4: $\quad P \leftarrow P \cup P_{\text{new}}$
5: $\quad$ Evaluate fitness $f(T)$ for each tree $T \in P$
6: $\quad P \leftarrow$ SelectBest($P, p$) $\qquad\qquad\qquad\qquad\qquad\qquad\qquad$ ▷ Keep top $p$ trees
7: **end for**
8: $T_{\text{best}} \leftarrow$ BestTree($P$)
9: **return** $T_{\text{best}}$

---

#### 3.3.2 ONE-SHOT SETTING

In the most extreme case, we only have one image per class. Thus, proposed methods involving K-fold validation will not be able to span the full class range of the dataset (Naghizadeh et al., 2021). To address this problem, we devised the following strategies:

---

**Algorithm 2** K-Fold Cross Validation Tree Fitness Function

---

**Require:** Dataset $D$, augmentation tree $T$, number of folds $k$
1: Split $D$ into $k$ folds: $D_1, D_2, \ldots, D_k$
2: Initialize $M \leftarrow 0$
3: **for** $i = 1$ to $k$ **do**
4:     $D_{\text{val}} \leftarrow D_i$
5:     $D_{\text{train}} \leftarrow D \setminus D_i$
6:     $D_{\text{aug}} \leftarrow \text{ApplyAugmentationTree}(T, D_{\text{train}})$
7:     Train model $M_i$ on $D_{\text{aug}}$
8:     $m_i \leftarrow \text{Evaluate}(M_i, D_{\text{val}})$
9:     $M \leftarrow M + m_i$
10: **end for**
11: $\bar{m} \leftarrow \frac{M}{k}$
12: **return** $\bar{m}$

---

**Label-Efficient Clustering** Our goal is to find augmentations that preserve important class-specific characteristics while still providing novel data. Thus, when evaluating on a validation set is not possible, we can switch to a clustering approach. To find these novel, true-to-class augmentations, our intuition is to search for clusters that are wide, but still distinct from each other. Abavisani et al. proposed using this type of evaluation for augmentation pipelines in low-data and one-shot settings (Abavisani et al., 2020). They adopted Deep Subspace Clustering (Ji et al., 2017) and optimized the Silhouette coefficient as a measure of cluster quality. We improve upon this work in three ways:

1. We simplify the clustering process by using a pre-trained network to generate image embeddings which we then cluster, thus eliminating the need for a Deep Subspace Clustering network and requiring no training.

2. Prior work employed k-means to form clusters (Douzas et al., 2018), adding computational complexity. We simplify this by directly using known class labels as clusters. This allows us to evaluate explicitly whether augmentations form meaningful, class-based clusters rather than merely measuring separability.

3. When evaluating augmentation quality via clustering, traditional metrics like the Silhouette coefficient reward cohesion but do not penalize small or redundant clusters. This can cause the evolutionary algorithm to favor augmentation trees that produce minimal or trivial variations, which lack diversity and generalization potential. To avoid this pitfall, we introduce an additional penalty term based on average cluster radius, balancing cohesion with cluster size and separability. This modified metric thus encourages the formation of clusters that are both cohesive and sufficiently distinct, promoting better generalization. Experiments supporting these conclusions are presented in Appendix A.5.

This process is given in Algorithm 3. The pipeline can be seen in figure 3b.

**Double Augmentation** This strategy is simple yet effective. We apply classical augmentations—which reliably introduce meaningful variations—to expand the original one-shot dataset. The augmented dataset is then divided into k splits, and the negative validation losses are averaged across splits, as detailed in Algorithm 4 and illustrated in Figure 3c. This approach allows us to increase augmentations while minimizing the risk of degrading dataset quality or relevance through unintended variations introduced by generative models.

---

**Algorithm 4** 1-Shot Double Augmentation Fitness Function

---

**Require:** One-shot dataset $D$, augmentation tree $T$, number of folds $k$
1: $D' \leftarrow \emptyset$
2: **for** each image $x \in D$ **do**
3:     $A(x) = \{\text{ClassicAug}(x)_1, \ldots, \text{ClassicAug}(x)_k\}$
4:     $D' \leftarrow D' \cup A(x)$
5: **end for**
6: **return** KFOLDFITNESS($D', T, k$)                    ▷ Refer to Alg. 2

---

---

**Algorithm 3** 1-Shot Clustering Fitness Function

---

**Require:** Image dataset $D$, augmentation tree $T$, embedding model $E$
 1: $D_{\text{aug}} \leftarrow \text{ApplyAugmentationTree}(T, D)$
 2: Initialize embedding list $L \leftarrow \emptyset$
 3: **for** each image $x \in D_{\text{aug}}$ **do**
 4:      $e \leftarrow E(x)$
 5:      Append $e$ to $L$
 6: **end for**
 7: $C \leftarrow \text{Cluster}(L)$
 8: $S \leftarrow \text{ComputeSilhouetteScore}(C)$
 9: $d \leftarrow \text{ComputeMeanClusterDistance}(C)$
10: $s \leftarrow \alpha S - \frac{1-\alpha}{d}$
11: **return** $s$

---

**Training Loss** We can also simply use training loss as a proxy in the one-shot case. We augment all the images, and train a model. We then evaluate trees based on how low the training loss is after a fixed number of epochs. While this should encourage minor augmentations, and also makes use of train loss to estimate eval loss, a very erroneous assumption, it still works well in practice. The pipeline can be seen in figure 3d.

## 4 RESULTS

### 4.1 EXPERIMENT SETUP

We perform our experiments on six datasets: Caltech256 (Griffin et al., 2007), Oxford IIIT-Pets (Parkhi et al., 2012), Oxford 102 Flowers (Nilsback & Zisserman, 2008), Stanford Cars (Krause et al., 2013), Stanford Dogs (Khosla et al., 2011), and Food101 (Bossard et al., 2014). To highlight how powerful our method is, even in few-shot settings when the fine-grained semantic distinctions are minor, we *deliberately searched* for few-shot images and classes that were the most challenging.

For an $n$-way $k$-shot classification task, we do this as follows. First, we randomly select $n$ classes from the original dataset. Then we randomly selected $k$ images from each class. We fine-tune a pretrained Resnet50 model (He et al., 2016) on these images and record the accuracy. We repeat this procedure 10 times, gathering 10 different subsets of the classes with different images for each dataset. Afterwards, we note which subset of classes from the dataset had the lowest baseline test accuracy, and we choose this subset as the setting for our augmentation benchmarks.

For our genetic algorithm, we initialize a population of 14. For each of the seven augmentation operators, we initialize two trees whose root nodes use that operator, creating a balanced population. This broadens the solution space exploration and avoids the pitfalls of random initialization on a small population. We set the mutation probability to $10\%$ and include 6 crossovers per generation. We restrict tree depth to 2, allowing the composition of at most 2 operations per augmentation. For each of the 10 generations, we generate 8 children. In the 2 and 5-shot cases, we use K-fold fitness, choosing folds such that the classes remained balanced. To evaluate augmentation trees, we train the models for 20 epochs and observe the corresponding loss. In the one-shot case, we examine the three other fitness functions (double augmentation, training loss, and clustering) proposed above.

Once the best tree is chosen, we generate augmentations and evaluate the downstream classification accuracy against several baselines:

1. Naive Baseline: We randomly apply classical augmentations (cropping, scaling, translation, horizontal/vertical flipping, color jitter, rotation)

2. RandAugment: We perform a grid search over the number of operations (num_ops) and magnitude parameters, selecting the configuration with the lowest validation loss using cross-validation on a train/validation split; this best-performing configuration is then evaluated on the full test set.

3. AutoAugment: We apply the ImageNet-learned AutoAugment policy to our datasets.

In all downstream classification tasks, training proceeds for 200 epochs. In the 1-shot setting, we augment each image in the original dataset 2 times, in the 2-shot setting 5 times, and in the 5-shot setting 2 times. We also evaluate our methods and baselines against augmentations generated from random trees in the ResNet experiments to ensure that our evolutionary search was an important part of creating true-to-class augmentations. Each experiment is performed at least three times with varying seeds, and the average and standard deviation are reported. We evaluate using a pre-trained ResNet50, ViT-Small (Dosovitskiy et al., 2021), or MobileNetV2 (Sandler et al., 2019) model. Models are fine-tuned using Adam (Kingma & Ba, 2017), with a learning rate of 1e-3. We use NVIDIA GeForce RTX 4090 chips with 24 GB of memory. Each experiment took between 2 and 24 hours to complete, depending on the number of ways and shots.

## 4.2 FEW-SHOT RESULTS

| Dataset | Model | Naive Baseline | Random Tree | RandAugment | AutoAugment | Learned |
|---|---|---|---|---|---|---|
| Caltech256 | ResNet50 | 79.78 ± 0.73 | 81.42 ± 7.64 | 84.71 ± 0.01 | 86.78 ± 0.00 | **88.28 ± 1.75** |
| | MobileNet | 80.95 ± 1.08 | - | **86.04 ± 0.01** | 84.45 ± 0.01 | 84.18 ± 0.75 |
| | ViT-Small | 73.30 ± 6.61 | - | 81.85 ± 0.01 | **81.74 ± 0.02** | 79.83 ± 9.90 |
| Flowers102 | ResNet50 | 70.49 ± 1.08 | 78.59 ± 2.11 | 83.65 ± 0.00 | **86.60 ± 0.00** | 73.73 ± 3.70 |
| | MobileNet | 77.00 ± 1.83 | - | **79.54 ± 0.01** | 81.75 ± 0.01 | 73.21 ± 4.75 |
| | ViT-Small | 97.89 ± 1.43 | - | **99.37 ± 0.00** | 98.63 ± 0.00 | 94.30 ± 3.85 |
| Stanford Dogs | ResNet50 | 78.44 ± 0.13 | 83.76 ± 6.42 | 80.76 ± 0.01 | 82.23 ± 0.00 | **85.15 ± 2.73** |
| | MobileNet | 75.34 ± 0.99 | - | 73.72 ± 0.02 | 75.26 ± 0.00 | **77.51 ± 0.63** |
| | ViT-Small | 83.67 ± 1.38 | - | **86.95 ± 0.02** | 83.97 ± 0.02 | 80.61 ± 2.01 |
| Stanford Cars | ResNet50 | 30.90 ± 1.68 | 36.94 ± 6.48 | 37.20 ± 0.01 | 34.68 ± 0.01 | **40.40 ± 3.07** |
| | MobileNet | 35.35 ± 1.05 | - | 36.30 ± 0.01 | 36.95 ± 0.01 | **37.36 ± 0.15** |
| | ViT-Small | 40.64 ± 5.43 | - | 43.83 ± 0.01 | 42.74 ± 0.02 | **46.32 ± 1.43** |
| Oxford-IIIT Pet | ResNet50 | 86.57 ± 0.60 | 84.97 ± 3.08 | 85.25 ± 0.01 | 86.57 ± 0.00 | **88.34 ± 1.72** |
| | MobileNet | 84.41 ± 0.53 | - | 87.27 ± 0.01 | 86.08 ± 0.00 | **89.21 ± 2.93** |
| | ViT-Small | 88.52 ± 0.55 | - | 91.28 ± 0.01 | 90.60 ± 0.01 | **91.44 ± 2.06** |
| Food101 | ResNet50 | 47.82 ± 0.57 | 42.61 ± 4.56 | 46.23 ± 0.00 | **51.32 ± 0.00** | 49.78 ± 4.21 |
| | MobileNet | 39.93 ± 1.46 | - | **43.49 ± 0.00** | 44.82 ± 0.00 | 42.97 ± 2.61 |
| | ViT-Small | 55.66 ± 3.22 | - | 62.20 ± 0.02 | 64.14 ± 0.02 | **64.18 ± 2.13** |

Table 1: 5-way, 2-shot classification accuracy (%) with standard deviation across 6 datasets and 3 downstream image classification architectures. Bolded values indicate the best performance per row.

| Dataset | Model | Naive Baseline | Random Tree | RandAugment | AutoAugment | Learned |
|---|---|---|---|---|---|---|
| Caltech256 | ResNet50 | 88.15 ± 0.25 | 92.22 ± 0.93 | 92.55 ± 0.00 | **93.31 ± 0.00** | 91.48 ± 1.11 |
| | MobileNet | 88.80 ± 0.19 | - | 89.51 ± 0.00 | **91.41 ± 0.00** | 90.05 ± 0.98 |
| | ViT-Small | 85.75 ± 1.04 | - | 90.32 ± 0.01 | 90.70 ± 0.01 | **92.33 ± 1.27** |
| Flowers102 | ResNet50 | 82.61 ± 0.96 | 79.51 ± 2.69 | 89.04 ± 0.01 | **89.92 ± 0.01** | 84.95 ± 1.15 |
| | MobileNet | 88.82 ± 0.69 | - | **91.47 ± 0.00** | 89.26 ± 0.01 | 86.60 ± 1.07 |
| | ViT-Small | 99.78 ± 0.19 | - | 99.89 ± 0.00 | **99.89 ± 0.00** | 99.34 ± 0.33 |
| Stanford Dogs | ResNet50 | 88.69 ± 0.65 | 90.81 ± 1.08 | 89.21 ± 0.00 | 91.24 ± 0.00 | **91.35 ± 0.72** |
| | MobileNet | 82.88 ± 0.72 | - | 81.49 ± 0.00 | 83.10 ± 0.00 | **83.40 ± 0.27** |
| | ViT-Small | 88.73 ± 0.46 | - | **89.73 ± 0.00** | 87.99 ± 0.00 | 84.66 ± 0.85 |
| Stanford Cars | ResNet50 | 52.97 ± 0.80 | 53.75 ± 1.44 | 54.63 ± 0.00 | 51.48 ± 0.01 | **57.98 ± 3.20** |
| | MobileNet | 50.79 ± 1.39 | - | 55.41 ± 0.01 | **55.93 ± 0.01** | 48.87 ± 1.71 |
| | ViT-Small | 58.12 ± 2.73 | - | 63.00 ± 0.02 | **66.67 ± 0.01** | 59.25 ± 2.23 |
| Oxford-IIIT Pet | ResNet50 | 92.07 ± 0.44 | 93.12 ± 0.74 | 92.91 ± 0.01 | 93.61 ± 0.00 | **93.63 ± 0.43** |
| | MobileNet | 88.56 ± 0.85 | - | 90.32 ± 0.00 | 90.14 ± 0.00 | **90.53 ± 0.76** |
| | ViT-Small | 94.95 ± 0.56 | - | **95.44 ± 0.01** | 95.37 ± 0.01 | 93.54 ± 0.53 |
| Food101 | ResNet50 | 54.09 ± 0.58 | 56.88 ± 2.60 | 56.72 ± 0.00 | 58.28 ± 0.00 | **58.75 ± 1.60** |
| | MobileNet | 51.88 ± 0.59 | - | 52.00 ± 0.00 | 54.05 ± 0.00 | **54.19 ± 1.36** |
| | ViT-Small | 75.40 ± 2.22 | - | **79.59 ± 0.00** | 78.25 ± 0.00 | 76.49 ± 0.75 |

Table 2: 5-way, 5-shot classification accuracy (%) with standard deviation across 6 datasets and 3 downstream image classification architectures. Bolded values indicate the best performance per row.

The few-shot results are shown in Tables 1 and 2. We measure the accuracy on the test set for models trained using the baseline strategies, random augmentation trees, and the augmentation trees learned from our pipeline. While EvoAug consistently outperforms the Naive Baseline, results are mixed when evaluated against AutoAugment and RandAugment. Notably, EvoAug is much better on the Stanford Dogs and Oxford-IIIT Pets datasets, but marginally worse on Flowers102.

| Dataset | Model | Naive Baseline | NoOp / Classical Tree | Random Tree | RandAugment | AutoAugment | Learned (Clustering) |
|---|---|---|---|---|---|---|---|
| Caltech256 | ResNet50 | 65.77 ± 1.29 | 78.67 ± 2.00 | 81.57 ± 6.44 | 81.63 ± 0.01 | 82.92 ± 0.01 | **83.65 ± 4.92** |
| | MobileNet | 67.28 ± 2.56 | - | - | 71.97 ± 0.01 | 71.47 ± 0.01 | **80.09 ± 4.73** |
| | ViT-Small | 66.72 ± 3.53 | - | - | 75.60 ± 0.03 | 75.38 ± 0.03 | **82.12 ± 3.64** |
| Flowers102 | ResNet50 | 61.48 ± 0.78 | 66.15 ± 0.90 | 63.03 ± 3.95 | 63.97 ± 0.00 | 65.84 ± 0.01 | **66.75 ± 2.34** |
| | MobileNet | 57.11 ± 1.30 | - | - | 53.17 ± 0.01 | 58.46 ± 0.01 | **60.78 ± 2.58** |
| | ViT-Small | 94.60 ± 1.88 | - | - | **96.26 ± 0.03** | 93.98 ± 0.02 | 95.47 ± 2.19 |
| Stanford Dogs | ResNet50 | 70.30 ± 0.58 | 75.79 ± 0.29 | 76.58 ± 3.84 | 75.86 ± 0.01 | 77.22 ± 0.02 | **78.86 ± 3.21** |
| | MobileNet | 60.58 ± 2.66 | - | - | 65.19 ± 0.02 | 67.73 ± 0.01 | **69.70 ± 2.37** |
| | ViT-Small | 75.55 ± 1.61 | - | - | 78.47 ± 0.01 | 77.83 ± 0.02 | **79.70 ± 3.12** |
| Stanford Cars | ResNet50 | 21.31 ± 0.80 | 28.11 ± 0.43 | 29.77 ± 1.83 | **32.84 ± 0.01** | 31.84 ± 0.03 | 29.66 ± 2.62 |
| | MobileNet | 30.43 ± 1.12 | - | - | 30.18 ± 0.01 | **30.35 ± 0.02** | 29.05 ± 3.18 |
| | ViT-Small | 31.10 ± 5.56 | - | - | 36.15 ± 0.02 | 34.66 ± 0.01 | **37.35 ± 3.37** |
| Oxford-IIIT Pet | ResNet50 | 79.68 ± 1.50 | 82.44 ± 0.55 | 81.47 ± 6.34 | 78.17 ± 0.01 | 82.71 ± 0.00 | **86.16 ± 1.19** |
| | MobileNet | 72.18 ± 1.45 | - | - | 76.31 ± 0.00 | 74.17 ± 0.01 | **80.43 ± 1.87** |
| | ViT-Small | 76.10 ± 5.44 | - | - | 83.88 ± 0.02 | 79.61 ± 0.04 | **84.58 ± 3.22** |
| Food101 | ResNet50 | 30.90 ± 0.57 | 30.06 ± 0.31 | 32.83 ± 2.42 | 30.38 ± 0.00 | 30.46 ± 0.00 | **34.28 ± 0.83** |
| | MobileNet | 28.78 ± 0.63 | - | - | 25.52 ± 0.01 | 26.15 ± 0.01 | **34.61 ± 1.74** |
| | ViT-Small | 43.74 ± 2.51 | - | - | **48.17 ± 0.02** | 45.44 ± 0.01 | 44.89 ± 2.30 |

Table 3: 5-way, 1-shot classification accuracy (%) with standard deviation across 6 datasets and 3 downstream image classification architectures. Bolded values indicate the best performance per row.

## 4.3 ONE-SHOT RESULTS

Our 1-shot results are shown in Table 3. Here, we include our clustering-based fitness function learning strategy. Results for our double augmentation and training loss strategies are included in the appendix. EvoAug consistently outperforms the Naive Baseline, and often outperforms RandAugment and AutoAugment, achieving strong performance in scarce data settings. We also run our pipeline restricting nodes to just classical or NoOp transformations and find that these restricted trees perform worse than our normal trees. This supports the conclusion that generative augmentation operators are an important part of performance.

## 5 CONCLUSION

We present an automated augmentation strategy that leverages advanced generative models, specifically controlled diffusion and NeRF operators, in combination with classical augmentation techniques. By employing an evolutionary search framework, our method automatically discovers task-specific augmentation policies that significantly improve performance in fine-grained few-shot and one-shot classification tasks. Experimental results on a diverse set of datasets demonstrate that our approach not only outperforms standard baselines but also identifies augmentation strategies that effectively preserve subtle semantic details, which are crucial in low-data scenarios.

Our work introduces novel unsupervised evaluation metrics and proxy objectives to reliably guide augmentation policy search in settings where labeled data is scarce. While the computational overhead associated with evaluating complex generative augmentations remains a challenge, the substantial gains in classification accuracy validate the potential of our approach. Overall, our findings suggest that integrating generative models with automated policy learning can play a pivotal role in enhancing the robustness of vision systems, particularly in environments with limited data.

### 5.1 LIMITATIONS

A potential limitation of our method is its ability to extend to a full dataset recognition task, as directly scaling our pipeline to learn semantic priors from the full dataset is not efficient. Preliminary work, however, has shown that using a text conditioned process to augment images does improve the performance of models on image classification tasks against a classical augmentation baseline (discussion in Appendix A.6). We believe that a more careful augmentation learning strategy that efficiently learns augmentations that match the dataset may be able to further improve this accuracy.

Other avenues of interest are extending this framework to other vision tasks such as object detection and segmentation and further refining the balance between diversity and fidelity in generated augmentations. Preliminary work on these tasks has shown that our pipeline has the ability to improve model performance when compared to a baseline of classically augmented images (discussion in Appendix A.6).

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

## A APPENDIX

We provide additional results for our method in the 5-way 1-shot setting, as well as a study on the one-shot clustering fitness function. We also examine how our method might scale to be used on full datasets and object detection/segmentation tasks.

## A.1 Fitness function choice in one-shot setting

Carefully crafting a fitness function which can enable robust downstream classification is a difficult task. Our three main approaches were using augmented images themselves as a part of the validation set for models, using heuristics from the training loss to determine optimal learning, and an involved clustering approach which tried to capture the spread within a class and between classes. Our results are summarized in Table 4, which show that the clustering approach seemed to be a consistently good strategy for guiding our augmentation scoring.

| Dataset | Model | Learned (Double Aug) | Learned (Train Loss) | Learned (Clustering) |
|---|---|---|---|---|
| Caltech256 | ResNet50 | 80.27 ± 6.50 | 76.21 ± 5.58 | **83.65 ± 4.92** |
| | MobileNet | 75.04 ± 6.68 | 68.57 ± 2.85 | **80.09 ± 4.73** |
| | ViT-Small | 73.53 ± 4.31 | 68.57 ± 10.44 | **82.12 ± 3.64** |
| Flowers102 | ResNet50 | 55.95 ± 5.72 | 65.73 ± 6.54 | **66.75 ± 2.34** |
| | MobileNet | 57.70 ± 1.45 | **64.38 ± 2.99** | 60.78 ± 2.58 |
| | ViT-Small | 88.27 ± 2.42 | 86.11 ± 1.85 | **95.47 ± 2.19** |
| Stanford Dogs | ResNet50 | 78.46 ± 2.27 | 68.26 ± 2.19 | **78.86 ± 3.21** |
| | MobileNet | 66.40 ± 2.56 | 67.25 ± 2.49 | **69.70 ± 2.37** |
| | ViT-Small | 74.73 ± 4.08 | 67.57 ± 1.55 | **79.70 ± 3.12** |
| Stanford Cars | ResNet50 | 22.34 ± 2.92 | 28.36 ± 0.86 | **29.66 ± 2.62** |
| | MobileNet | 25.29 ± 4.40 | 26.70 ± 1.88 | **29.05 ± 3.18** |
| | ViT-Small | 24.79 ± 0.63 | 36.73 ± 0.80 | **37.35 ± 3.37** |
| Oxford-IIIT Pet | ResNet50 | 76.07 ± 2.26 | 78.30 ± 1.44 | **86.16 ± 1.19** |
| | MobileNet | 75.76 ± 2.58 | 74.45 ± 4.54 | **80.43 ± 1.87** |
| | ViT-Small | 75.69 ± 4.36 | 80.58 ± 3.62 | **84.58 ± 3.22** |
| Food101 | ResNet50 | 30.40 ± 1.56 | 30.49 ± 3.22 | **34.28 ± 0.83** |
| | MobileNet | 28.44 ± 0.89 | 29.38 ± 2.32 | **34.61 ± 1.74** |
| | ViT-Small | 38.17 ± 1.53 | 39.53 ± 4.35 | **44.89 ± 2.30** |

Table 4: 5-way, 1-shot classification accuracy (%) with standard deviation across 6 datasets and 3 downstream architectures, showing only the three Learned methods. Bolded values indicate the best performance per row.

## A.2 Encoder Performance Comparison

The one-shot clustering fitness function results only use a single image encoder, a pre-trained ResNet50. We begin this analysis by benchmarking various pre-trained image encoders—responsible for projecting augmented images into embedding space—for their effectiveness in the clustering-based fitness function. We explore two variants of Vision Transformers (Dosovitskiy et al., 2021) in addition to a ResNet50. Table 5 provides the results of the encoder performance comparison. The two vision transformer variants outperform the baseline on all datasets. Notably, however, there is no single best decoder that performs consistently the best across all datasets.

Table 5: Accuracy for 5-way 1-shot clustering fitness function across various image encoders

| Dataset | Baseline | ResNet50 | ViT-224 | ViT-B/16 |
|---|---|---|---|---|
| Caltech256 | 65.77 ± 1.29 | **81.83 ± 7.60** | 79.56 ± 1.10 | 72.28 ± 1.98 |
| Flowers102 | 61.48 ± 0.78 | 56.70 ± 4.19 | 62.41 ± 2.38 | **64.38 ± 3.30** |
| Stanford Dogs | 70.30 ± 0.58 | 71.54 ± 2.70 | **75.45 ± 2.25** | 72.72 ± 2.42 |
| Stanford Cars | 21.31 ± 0.80 | 24.50 ± 3.00 | **25.71 ± 2.74** | 24.38 ± 1.32 |
| Oxford-IIIT Pet | 79.68 ± 1.50 | 85.21 ± 1.14 | 83.08 ± 2.81 | **85.88 ± 0.66** |
| Food101 | 30.90 ± 0.57 | **33.97 ± 1.63** | 32.21 ± 0.41 | 32.62 ± 0.67 |

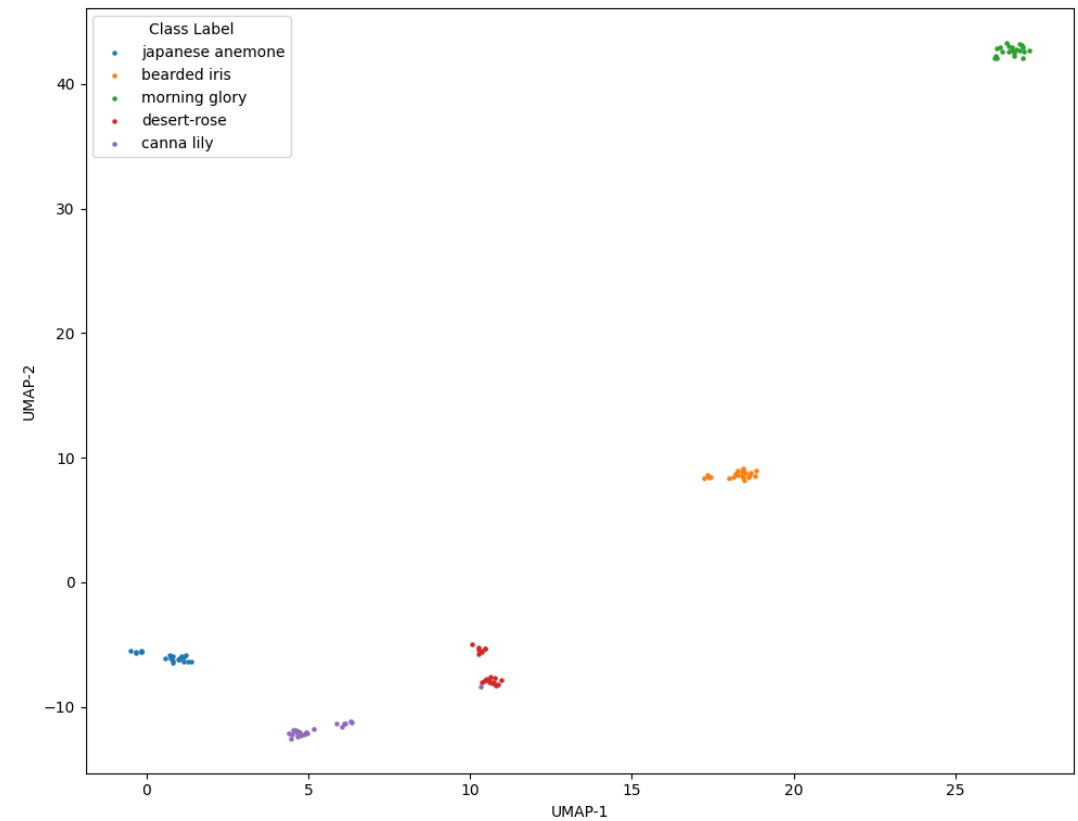

Figure 4: Success Case: ViT-B/16-Flowers

### A.3    SUCCESS AND FAILURE ANALYSIS

We look at low-dimensional cluster visualizations of encoded augmentations from strong and weak-performing learned trees for success and failure cases using UMAP (McInnes et al., 2018). This motivates the desired and non-desired qualities of clusters. We examine the clusters of the embeddings of two encoders on the Flower dataset, shown in Figure 4 and Figure 5. We select a single dataset to establish domain consistency when comparing success and failure cases, as well as against the handcrafted tree study in the following section. The Flowers102 dataset is particularly interesting as it is the most fine-grained among those benchmarked. Unlike other datasets, where shape or size may be primary distinguishing features between classes, flowers are primarily defined by their color. As a result, applying augmentations that alter color can significantly degrade model performance.

For the success case – ViT-B/16 on Flowers102 – which performed 3% better than baseline, there are distinct clusters for all five classes, all of which are very tight. Clusters are also very well separated. For the failure case – ResNet50 on Flowers102 – which performed 5% worse than baseline, the classes are not clustered very accurately, with augmentations overlapping heavily between classes.

### A.4    HANDCRAFTED AUGMENTATION TREES

We handcraft an "ideal" augmentation tree for the Flowers102 dataset, shown in Table 6, to compare to the clusters of the EvoAug learned trees in the success and failure cases. The hypothesized ideal augmentation tree is structured as follows: the head node as Color, the left node as NeRF, and the right node as no augmentation, with a 0.5 probability of moving to either child node. We guarantee a Color node, as it uses Color ControlNet to preserve the color palette in augmentations. We also use a NeRF node, which performs a 3D rotation for an augmentation, yet not affecting color.

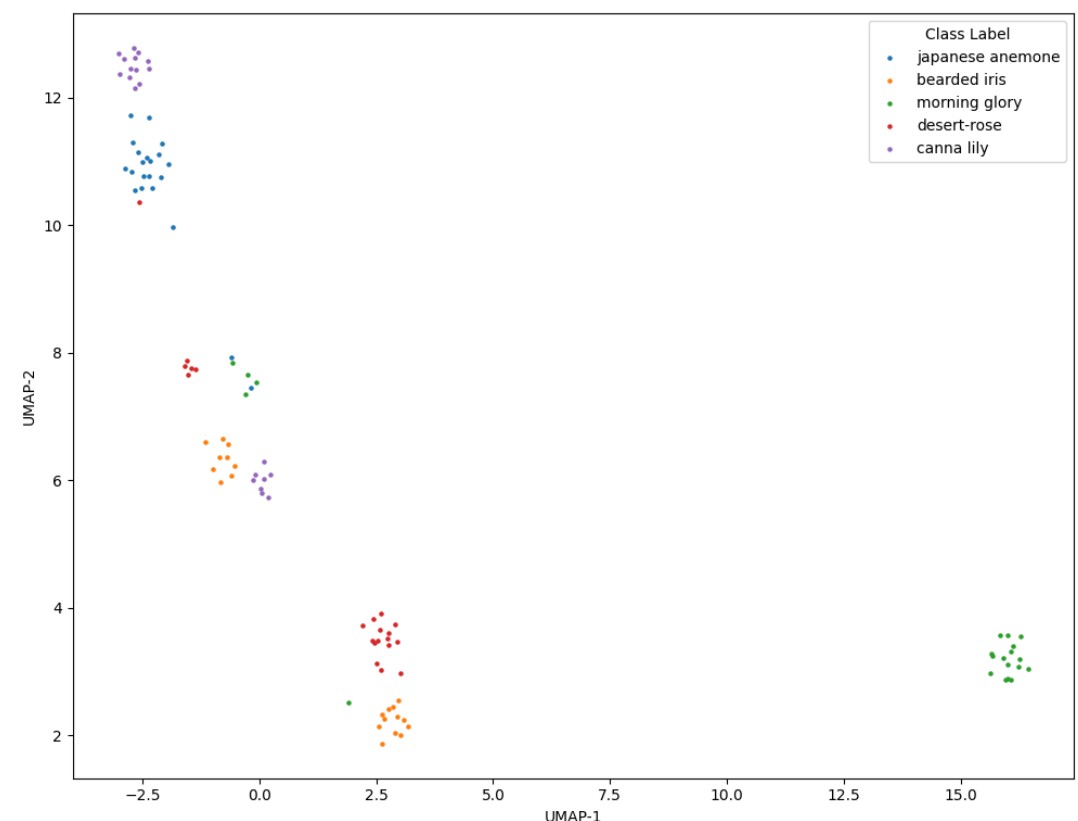

Figure 5: Failure Case: ResNet50-Flowers

Table 6: Handcrafted tree performance on the Flowers102 dataset. Tree structure format: (Head, $p_L$, Left, $p_R$, Right).

| Name | Tree Structure | Accuracy (%) |
|---|---|---|
| Ideal | (Color, 0.5, NeRF, 0.5, None) | $66.98 \pm 6.56$ |
| Inferior | (Depth, 0.5, Depth, 0.5, Segmentation) | $60.85 \pm 2.30$ |

We also handcraft an "inferior" augmentation tree as a sanity check and counterexample, allowing us to compare clusters and better isolate critical features to reward when designing the clustering fitness score. We use Depth and Segmentation nodes for augmentations, as neither augmentation operation preserves color, which we hypothesize to be the most important feature for flower classification.

The handcrafted ideal augmentation tree performs better than all other augmentation trees learned from any image encoder, suggesting that the EvoAug pipeline is not learning the best augmentation tree through the clustering score fitness function. The ideal handcrafted tree in Figure 6 and the learned tree success case in Figure 4 both display very well-separated clusters for each class. However, the clusters for the success case are noticeably tighter than those of the handcrafted tree

| Image Encoder | $S - \frac{1}{d}$ | $S - \frac{2}{d}$ | $S$ | $\frac{1}{DB}$ |
|---|---|---|---|---|
| ViT-B/16 | $64.382 \pm 3.302$ | $67.497 \pm 2.827$ | $61.059 \pm 0.44$ | $61.059 \pm 0.44$ |

Table 7: One-shot clustering results across different fitness functions for Flowers102 subset 50

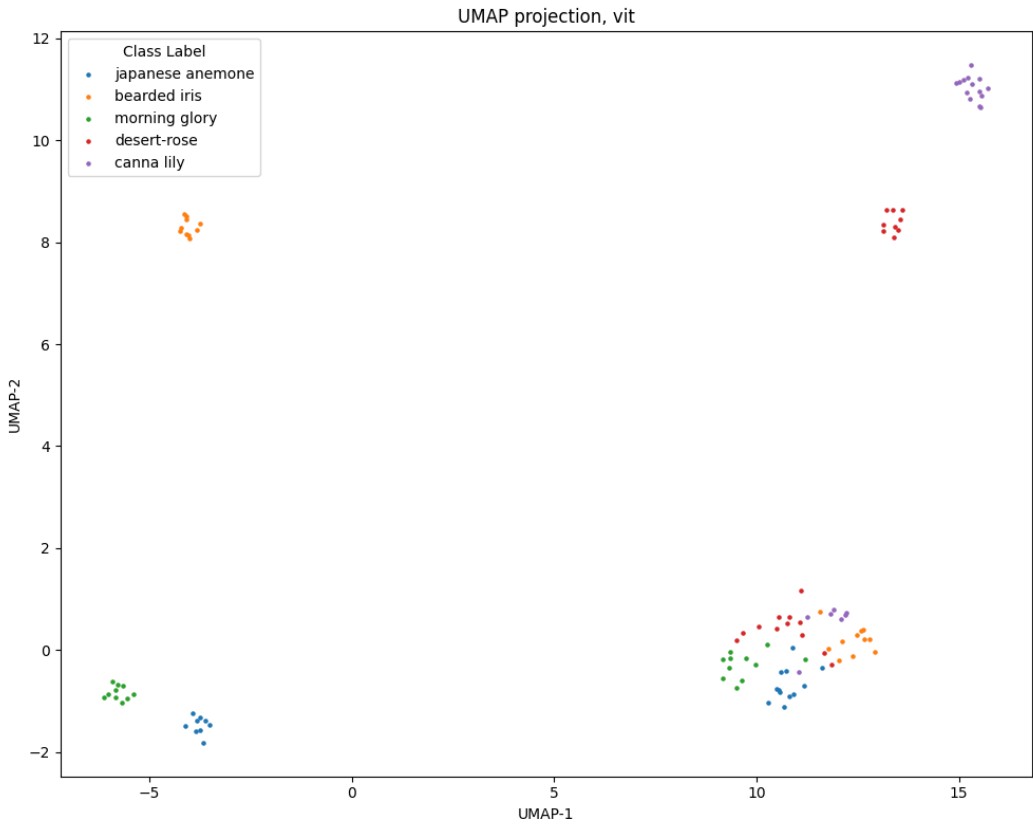

Figure 6: Handcrafted Ideal Tree

clusters. If we compare this to Figure 5 or 7, we can see larger clusters formed from a variety of different classes, with fewer clusters that distinctly correspond to a single class.

These observations give rise to two interpretations: (1) the original fitness function may have undervalued the importance of large clusters, because the better performing handcrafted ideal tree resulted in larger yet still distinct clusters and (2) that the original fitness function may have overvalued the importance of large clusters at the expense of cluster separability, as the failure case and handcrafted inferior tree demonstrate. This motivates an exploration of alternative fitness functions that may better capture cluster dynamics.

## A.5 CLUSTERING FITNESS FUNCTION MODIFICATIONS

Table 7 compares the performance of different clustering metrics as the fitness function in the EvoAug pipeline, where $\mathbf{S}$ is the Silhouette coefficient, $\mathbf{d}$ is the average cluster radius, $\mathbf{DB}$ is the Davies-Bouldin Index (Davies & Bouldin, 1979). We conduct experiments using the Flowers102 dataset and use ViT-B/16 encoder as it performs the best on this dataset.

We test a fitness function of just $\mathbf{S}$ as a baseline, but using only the Silhouette Coefficient results in a learned tree of None nodes, causing all generated augmentations to be exact copies of the original image. This is expected, as the Silhouette Coefficient scores clusters of the same embedding as a perfect score of 1, due to the small intra-cluster distances. The same result occurs with the $\frac{1}{\mathbf{DB}}$ fitness function, confirming that Davies-Bouldin is functionally the same as the Silhouette Coefficient.

We modify the original proposed fitness function by doubling the penalty to small cluster sizes. Under this setting, the learned augmentation tree is $(\text{Head}, p_L, \text{Left}, p_R, \text{Right}) = (\text{None}, 0.51, \text{None}, 0.49, \text{NeRF})$. This tree results in the best downstream classification performance

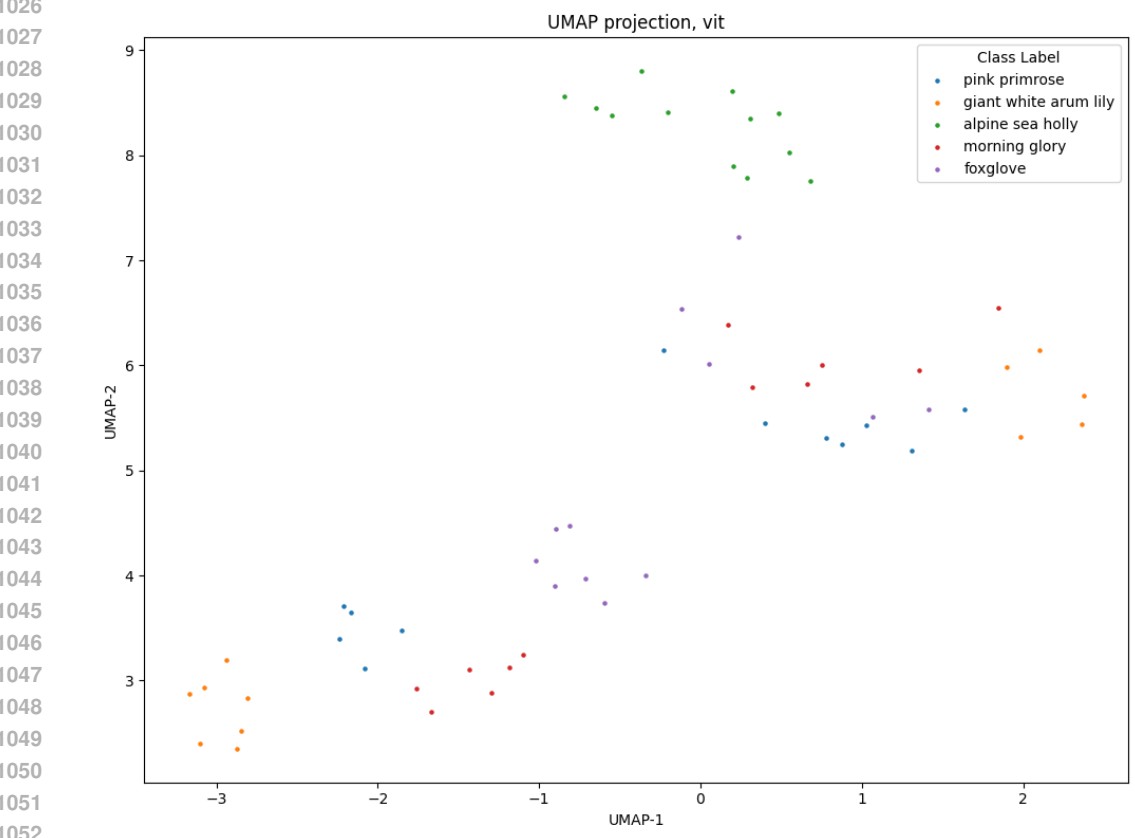

Figure 7: Handcrafted Inferior Tree

across all experiments, including those from handcrafted trees, demonstrating that this fitness function was able to learn better trees than human intuition. This learned tree was likely favored in the evolutionary algorithm, as NeRF preserves colors and edges, two features we believe are vital for classifying flowers. These results strengthen the interpretation that a large intra-cluster distance is important may help in model generalization. Future work will seek to substantiate this claim in other settings and datasets.

## A.6 GENERALIZATION TO FULL DATASETS, DETECTION, AND SEGMENTATION

While the main body of our work focuses on the few shot setting, there are also experiments done which have indicated that conditioned generation is beneficial in the full dataset setting (Anonymous, 2024). The method used in these experiments employs LLaVa2 (Liu et al., 2023a) generated captions to condition the augmentation of images in the dataset. We believe that with more intelligent conditioning (by learning augmentation trees which match the dataset), we can achieve better performance.

We reproduce the relevant summary statistics below in Table 8 for completeness. The results show that conditioned generation consistently achieves higher accuracy than a classically augmented baseline across six datasets: Caltech256 (Griffin et al., 2007), Stanford Cars (Krause et al., 2013), FGVC Aircraft (Maji et al., 2013), Stanford Dogs (Khosla et al., 2011), Oxford IIIT-Pets (Parkhi et al., 2012); and eight model architectures: ResNet (He et al., 2016), VGG (Simonyan & Zisserman, 2014), EfficientNet (Tan & Le, 2019), Visformer (Chen et al., 2021), Swin Transformer (Liu et al., 2021), MobileNet (Howard et al., 2017), DenseNet (Iandola et al., 2014), and ViT (Dosovitskiy et al., 2021).

We have also done some 5-way, 2-shot experiments on the PASCAL VOC dataset (Everingham et al., 2010). For these experiments, we fine-tuned a Faster R-CNN (Ren et al., 2015) with a ResNet-50-FPN backbone (Lin et al., 2017) pretrained on COCO (Lin et al., 2014). Our results show that a baseline strategy which only uses classical augmentations achieves a performance of $18.77 \pm 5.95$ percent, while our generative augmentation pipeline achieves a performance of $21.53 \pm 7.20$ percent. This indicates that our generative augmentation pipeline can also benefit dense prediction tasks.

Table 8: Accuracy on full datasets for various models

| Dataset | Setting | RN50 | RN101 | VGG19 | EN | Visformer | Swin | MN | DN |
|---------|---------|------|-------|-------|-----|-----------|------|-----|-----|
| Caltech | Baseline | 72.37 | 73.62 | 67.40 | 71.79 | 68.83 | 63.95 | 66.48 | 75.74 |
|         | Conditioned | **76.49** | **77.64** | **70.82** | **73.85** | **73.15** | **69.55** | **68.33** | **78.10** |
| Cars | Baseline | 86.78 | 88.16 | 87.22 | 86.75 | 83.37 | 75.43 | 80.80 | 91.08 |
|      | Conditioned | **91.02** | **90.95** | **89.61** | **88.56** | **87.40** | **82.32** | **82.70** | **92.20** |
| Aircraft | Baseline | 75.23 | 75.91 | **88.80** | 81.25 | 72.61 | 60.88 | 70.24 | 80.53 |
|          | Conditioned | **82.33** | **81.10** | 88.20 | **81.76** | **74.67** | **71.74** | **74.17** | **83.29** |
| Dogs | Baseline | 66.49 | 70.15 | **68.63** | **64.17** | **64.65** | 52.10 | **58.60** | **70.44** |
|      | Conditioned | **68.74** | **70.40** | 66.05 | 62.45 | 64.36 | **56.50** | 58.30 | 70.21 |
| Pets | Baseline | 69.22 | 70.72 | **83.17** | 73.59 | 73.02 | 58.54 | 67.35 | **80.16** |
|      | Conditioned | **71.07** | **74.03** | 81.28 | **74.41** | **76.24** | **61.00** | **68.46** | 79.34 |

