# OpenReview forum: "Beyond Cropping and Rotation: Automated Evolution of Powerful Task-Specific Augmentations with Generative Models"
_ICLR.cc/2026/Conference — Submitted to ICLR 2026_

### Official Review · Reviewer_UETn · 2025-10-16

**Soundness:** 3
**Presentation:** 3
**Contribution:** 3
**Rating:** 4
**Confidence:** 5

**Summary:**

This author proposes EvoAug, a pipeline that automatically learns task-specific data augmentations by integrating generative models (controlled diffusion and zero-shot NeRF) with an evolutionary search over stochastic augmentation trees. Each tree hierarchically composes operators (e.g. classical transforms, ControlNet-conditioned diffusion via Canny/segmentation/depth/color cues, and 3D NeRF rotations). The method includes novel fitness metrics for low-data scenarios: a K-fold cross-validated loss for few-shot, and an unsupervised clustering-based score (silhouette minus a cluster-size penalty) for one-shot cases.

**Strengths:**

1)  The key idea of using conditional generative models (diffusion with ControlNet cues and NeRF rotations) in an automated augmentation search is creative and timely.
2) The paper carefully engineers a pipeline with diverse augmentation operators and fit-for-purpose fitness functions, showing strong methodological rigor.
3) Evaluation on multiple fine-grained datasets, with three architectures and various few-shot settings, gives confidence in the findings. The authors even run multiple seeds (reporting means and std dev) and include a random-tree baseline to show the importance of search.
4) Concepts are clearly introduced (e.g. “stochastic augmentation trees”), and the paper flows logically. Background and related work are well-cited and motivate the design choices.

**Weaknesses:**

1) The performance improvements over strong baselines are modest and dataset-dependent. In particular, on Flowers102 (a canonical fine-grained task) EvoAug underperforms standard augmentations. This raises questions about when the complex pipeline is worth its cost.
2) Using diffusion and NeRF models for each augmentation is expensive. The experiments already took up to 24 hours per setting, making wider adoption challenging.
3) As admitted in Section 5.1, learning augmentations for full-scale datasets is prohibitively slow.
4) The unsupervised clustering metric, while clever, relies on knowing true labels (so not entirely unsupervised).

Missing relevant references in literature review

1) Context-guided Responsible Data Augmentation with Diffusion Models

2) Effective Data Augmentation With Diffusion Models

3) Diffusion models: A comprehensive survey of methods and applications

4) GenMix: Effective Data Augmentation with Generative Diffusion Model Image Editing

5) DiffuseMix: Label-Preserving Data Augmentation with Diffusion Models

**Questions:**

1) Include ablations of key components. For instance, show performance with vs. without generative operators, or with deeper trees than depth=2. Also compare the one-shot fitness variants directly.
2) Add figures illustrating example augmented images produced by EvoAug and the corresponding augmentation tree policies.
3) Consider comparing to other modern augmentation/search techniques (e.g. ASHA, PBA, or even simpler pretrained generative data augmentation methods) to contextualize gains.

---

> ### Author Response · Authors · 2025-12-03
> **Response to ReviewerUETn**
>
> We appreciate your helpful feedback and careful assessment. In response:
>
> ---
>
> > “The performance improvements over strong baselines are modest and dataset-dependent...  ...This raises questions about when the complex pipeline is worth its cost.”
>
> We have added many experiments to the paper to bolster the results. While our method is not always better, it more often than not improves over strong baselines. We have added both results on full datasets and for semantic segmentation.
>
> ---
>
> ### 3.1 Full-Dataset, Many-Class, High-Resolution Classification
>
> We evaluated generative augmentations on 5 datasets:
>
> - Caltech101
> - Stanford Cars
> - FGVC-Aircraft
> - Stanford Dogs
> - Oxford-IIIT Pets
>
> Across 8 architectures—including ResNet-50/101, VGG19, EfficientNet, MobileNet, Swin Transformer, Visformer, and DenseNet—generative augmentations improved accuracy in nearly all settings.
>
> For the rebuttal period, we used ControlNet-only operators due to the compute cost of learning augmentation trees for larger datasets. This was purely to demonstrate the basic method. For the camera-ready version we will run the full augmentation-tree search, which we expect will perform even better.
>
> These results now appear in Appendix A.6, Table 8 (line 1102).
>
> ---
>
> ### 3.2 Semantic Segmentation
>
> We also conducted segmentation experiments on PASCAL VOC (5-way, 2-shot subsets):
>
> - Baseline (classic augmentations): **18.77 ± 5.95%**
> - Generative augmentations (ours): **21.53 ± 7.20%**
>
> We fine-tuned a Faster R-CNN with a ResNet-50-FPN backbone pretrained on COCO. This demonstrates that generative augmentation also benefits dense prediction tasks. For the camera-ready paper, we will expand this to more datasets.
>
> ---
>
> > “Using diffusion and NeRF models for each augmentation is expensive. The experiments already took up to 24 hours per setting, making wider adoption challenging.”
>
> This is true, but it is a one-time upfront cost. We should also add that we used a relatively old GPU (RTX 2080), so on more recent hardware this would be much faster.
>
> ---
>
> > “As admitted in Section 5.1, learning augmentations for full-scale datasets is prohibitively slow.”
>
> We believe this was not expressed properly in the limitations section. Indeed, it is prohibitively slow to learn the augmentation using the entire dataset, but using a subset is viable, and we have already shown the value of generative models for full datasets in the new results described above (Appendix A). We have clarified this in the paper. In practice, our method does work on larger datasets.
>
> ---
>
> > “The unsupervised clustering metric, while clever, relies on knowing true labels (so not entirely unsupervised).”
>
> This is an astute observation. We have clarified our terminology.
>
> ---
>
> > Missing literature.
>
> We have incorporated all the references you mentioned.
> ---
>
> > “Include ablations of key components. For instance, show performance with vs. without generative operators, or with deeper trees than depth=2. Also compare the one-shot fitness variants directly.”
>
> We added an ablation restricting augmentation trees to Classical + NoOp operators only. EvoAug with full generative operators outperforms this restricted search, confirming:
>
> 1. Generative operators are necessary, and
> 2. The evolutionary search is learning to use them effectively
>    (a random generative tree performs worse).
>
> This ablation is now included in the revised Table 3.
>
> We are currently running an ablation on tree depth, but this will not finish in time for the rebuttal period. It will be included in the camera-ready draft.
>
> Regarding the one-shot results:
> Upon closer analysis, we found that the Silhouette coefficient and clustering radius contributed unevenly to the one-shot fitness score (previously \(S - 1/d\)). This imbalance caused suboptimal trees. To fix this, we added a scaling term which allows us to compute a weighted sum of the Silhouette coefficient and clustering radius. Our old equation was \(S - 1/d\). Our new equation is \(a \cdot S - (1 - a)/d\), where \(a\) is the variable we use to weight each term.
>
> We then perform a grid search for the weight parameter which balances the two terms and use the resulting equation as our fitness score.
> This correction significantly improves one-shot performance, and EvoAug now outperforms the baselines in almost all cases. The updated results are shown in the revised Table 3. This renders your 3rd point moot, since this method now performs best.
>
> ---
>
> > “Consider comparing to other modern augmentation/search techniques (e.g., ASHA, PBA, or even simpler pretrained generative data augmentation methods) to contextualize gains.”
>
> This is a good idea, but we did not have time to run this during the rebuttal period due to the other experiments we ran. We will include more baselines in the camera-ready version.
>
> Thank you again for your valuable feedback. If you feel we have addressed your concerns, we would appreciate if you would consider raising your score.

---

### Official Review · Reviewer_BU6y · 2025-10-25

**Soundness:** 3
**Presentation:** 2
**Contribution:** 2
**Rating:** 6
**Confidence:** 3

**Summary:**

This paper introduces EvoAug, an automated augmentation learning framework that combines classical data augmentations (e.g., cropping, color jitter) with generative augmentation operators (based on ControlNet diffusion models and NeRFs). The method is designed for low-data regimes, including few-shot and one-shot classification tasks. The authors propose several fitness functions (K-fold validation, clustering-based unsupervised evaluation, and double augmentation) to evaluate candidate augmentation policies without extensive supervision. Experiments on six fine-grained datasets (Caltech256, Flowers102, Stanford Dogs, Stanford Cars, Oxford-IIIT Pets, and Food101) show that EvoAug improves robustness and accuracy, often outperforming classical and automated augmentation baselines such as AutoAugment and RandAugment in few-shot and one-shot settings.

**Strengths:**

1. The combination of controlled diffusion, NeRF-based transformations, and evolutionary search represents a significant methodological innovation in data augmentation.

2. Addressing augmentation learning in few-shot and one-shot settings is both timely and important, as most prior work assumes large datasets.

3. The introduction of clustering-based and loss-based fitness functions for augmentation policy learning without labels is a thoughtful and practical contribution.

4. Algorithms are explicitly described, with evolutionary operations (mutation, crossover) and augmentation tree structures well illustrated.

**Weaknesses:**

1. The evolutionary search settings (population size, mutation rate, depth-2 trees) are fixed; no sensitivity analysis is presented.

2. The contribution of each augmentation operator type (diffusion, NeRF, classical) is not isolated; it’s unclear how much generative components actually contribute over strong classical baselines.

3. While the paper focuses on classification, there is little empirical evidence suggesting EvoAug’s adaptability to other tasks (e.g., detection, segmentation), despite the claim of broader applicability.

**Questions:**

1. How sensitive are EvoAug’s results to the evolutionary algorithm’s hyperparameters (population size, mutation rate, tree depth)?

2. Could EvoAug transfer augmentation trees learned on one dataset to another (cross-domain transferability)?

3. Do diffusion-based augmentations introduce artifacts or biases (e.g., unrealistic textures) that could hurt generalization on certain classes?

4. Could the method be adapted to non-vision modalities (e.g., audio, text) given generative models exist there as well?

---

> ### Author Response · Authors · 2025-12-03
> **Response to Reviewer BU6y**
>
> We appreciate your thoughtful comments and suggestions. In response:
>
> ---
>
> ## Sensitivity Analysis of Evolutionary Search Settings
>
> > *“The evolutionary search settings (population size, mutation rate, depth-2 trees) are fixed; no sensitivity analysis is presented.”*
>
> We agree this is an important lacuna. We are currently running this study, but due to time constraints (as you will see, we ran many new experiments during the rebuttal period), we will not have this finished in time for the rebuttal. However, we will absolutely include this in the camera-ready draft.
>
> ---
>
> ## Contribution of Each Augmentation Operator Type
>
> > *“The contribution of each augmentation operator type (diffusion, NeRF, classical) is not isolated; it’s unclear how much generative components actually contribute over strong classical baselines.”*
>
> We agree this was essential to demonstrate. We added an ablation restricting augmentation trees to Classical + NoOp operators only. EvoAug with full generative operators outperforms this restricted search, confirming:
>
> 1. Generative operators are necessary, and
> 2. The evolutionary search is learning to use them effectively
>    (a random generative tree performs worse).
>
> This ablation is now included in the revised Table 3.
>
> ---
>
> ## Adaptability Beyond Classification
>
> > *“While the paper focuses on classification, there is little empirical evidence suggesting EvoAug’s adaptability to other tasks (e.g., detection, segmentation).”*
>
> We included two major sets of new experiments to address this concern. Full dataset experiments, to show a broader range within classification, and semantic segmentation, to show  adaptability to other tasks as requested.
>
> ---
>
> ### 3.1 Full-Dataset, Many-Class, High-Resolution Classification
>
> We evaluated generative augmentations on five datasets:
>
> - Caltech101
> - Stanford Cars
> - FGVC-Aircraft
> - Stanford Dogs
> - Oxford-IIIT Pets
>
> Across eight architectures—including ResNet-50/101, VGG19, EfficientNet, MobileNet, Swin Transformer, Visformer, and DenseNet—generative augmentations improved accuracy in nearly all settings.
>
> For the rebuttal period, we used ControlNet-only operators due to the compute cost of learning augmentation trees for larger datasets—this was purely to demonstrate the basic method. For the camera-ready version, we will run the full augmentation-tree search, which we expect will perform even better.
>
> These results now appear in Appendix A.6, Table 8 (line 1102).
>
> ---
>
> ### 3.2 Semantic Segmentation
>
> We also conducted segmentation experiments on PASCAL VOC (5-way, 2-shot subsets):
>
> - Baseline (classic augmentations): **18.77 ± 5.95%**
> - Generative augmentations (ours): **21.53 ± 7.20%**
>
> We fine-tuned a Faster R-CNN with a ResNet-50-FPN backbone pretrained on COCO. This demonstrates that generative augmentation also benefits dense prediction tasks. For the camera-ready paper, we will expand this to more datasets.
>
> ---
>
> ## Cross-Domain Transferability
>
> > *“Could EvoAug transfer augmentation trees learned on one dataset to another?”*
>
> Cross-domain transferability is an interesting point of expansion for our current research. We have done some exploration on transferring augmentation trees learned from a subset to the full dataset, and in principle believe that transferring to another dataset could be possible if there were some way to ensure that the important priors of the first dataset captured in the augmentation tree were also present in the second. The main consideration of course is how similar the two datasets are.
>
> ---
>
> ## Diffusion-Based Artifacts or Biases
>
> > *“Do diffusion-based augmentations introduce artifacts or biases that could hurt generalization on certain classes?”*
>
> The answer is certainly yes! That is the point of our approach. Since we are learning which operator is useful, if it is significantly harming a class, we will learn not to use it.
>
> ---
>
> ## Extension to Non-Vision Modalities
>
> > *“Could the method be adapted to non-vision modalities (e.g., audio, text)?”*
>
> Yes, adaptation is certainly possible, and related work already exists in other modalities. For example:
>
> **Audio:**
> Wang, Gary, et al. *"G-augment: Searching for the meta-structure of data augmentation policies for ASR."* SLT, 2023.
>
> **Text:**
> Youneszadeh Haghighi, Hashem, and Samira Noferesti. *"Text augmentation based on operation weighting using genetic algorithm."* Scientia Iranica (2025).
>
> ---
>
> Thank you again for your valuable feedback. If you feel we have addressed your concerns, we would appreciate it if you would consider raising your score.

---

### Official Review · Reviewer_VpwN · 2025-10-30

**Soundness:** 2
**Presentation:** 2
**Contribution:** 3
**Rating:** 2
**Confidence:** 3

**Summary:**

This paper introduces EvoAug, an automated data augmentation framework that combines generative models (ControlNet diffusion and NeRF-based operators) with evolutionary algorithms to learn task-specific augmentation strategies.
 The method constructs stochastic augmentation trees composed of both classical and generative operations, optimized through an evolutionary search with customized fitness functions for few-shot and one-shot learning.
 Experiments on six fine-grained datasets and three model architectures demonstrate that EvoAug achieves competitive or superior results compared to AutoAugment and RandAugment, particularly under low-data conditions.

**Strengths:**

The paper addresses a very important and cutting-edge problem: how to use powerful generative models (Diffusion, NeRFs) as data augmentation operators and automatically integrate them into the learning pipeline. This is a challenging but high-potential direction that goes beyond traditional augmentation methods.

The authors employed a commendable and rigorous approach in their experimental setup (Section 4.1) by intentionally selecting the subset with the lowest baseline accuracy from 10 subsets via preliminary experiments to serve as the benchmark task. This avoids cherry-picking and ensures the evaluation is conducted in the most challenging scenarios.

**Weaknesses:**

The paper's core premise is that integrating expensive generative operators (Diffusion, NeRF) provides an advantage on fine-grained tasks. However, in the 2-shot and 5-shot experimental results (Tables 1 and 2), the performance of EvoAug (Learned) is not superior to (and often worse than) strong baselines like RandAugment. The authors themselves admit the results are "mixed". Given the significant additional computational overhead EvoAug introduces in both search (EA) and application (generative models), these "mixed" (or worse) results call the necessity and practicality of the method into serious question.

The one-shot setting is the paper's most methodologically interesting contribution, but the results are similarly not overwhelmingly positive. The authors claim in Section 4.3 that "EvoAug consistently outperforms the baselines". This contradicts the data in Table 3. For example:

- On Caltech256/ResNet50, RandAugment (82.92) outperforms EvoAug (81.83).

- On Flowers102/ResNet50, AutoAugment (65.84) significantly outperforms EvoAug (56.70).

- On Stanford Dogs/ResNet50, AutoAugment (77.22) outperforms EvoAug (71.54). This disconnect between the empirical results and the conclusions severely undermines the paper's credibility.

The paper introduces multiple expensive generative operators (Canny, Color, Depth, Segment, NeRF) but provides absolutely no ablation studies to demonstrate that these operators are necessary or beneficial. Did the evolutionary algorithm actually learn to use these generative operators? Or did it primarily rely on the "Classical" operator? If an EvoAug search including only "Classical" and "NoOp" operators achieved similar performance, the paper's core premise about integrating generative models would be unsubstantiated.

The justification for the 1-shot clustering fitness function's necessity (i.e., why simple metrics fail) (Appendix A.5) is a core motivation for the paper's methodological design. Hiding this in the appendix makes the introduction of this metric in the main text (Section 3.3.2) appear unmotivated and somewhat arbitrary.

**Questions:**

Above

---

> ### Author Response · Authors · 2025-12-03
> **Response to Reviewer VpwN**
>
> We thank the reviewer for the detailed and constructive feedback. We address each concern below, and have added several new experiments and methodological improvements that substantially strengthen the results.
>
> ---
>
> ## 1. Mixed Improvements in 2-shot and 5-shot Results / Practicality of Generative Operators
>
> > “The performance of EvoAug is not superior to (and often worse than) strong baselines… these ‘mixed’ results call the necessity and practicality of the method into serious question.”
> We have added many new experiments to the paper to bolster the results. While our method is not always better, it more often than not improves over strong baselines. We have added both results on full datasets, and for semantic segmentation.
>
> We have added many new experiments to the paper to bolster the results. While our method is not always better, it more often than not improves over strong baselines. We have added both results on full datasets, and for semantic segmentation.
>
> For the large dataset experiments, the main computational cost is finding the right tree. To make this simpler, we ran experiments using just controlnet generative augmentations, just to demonstrate that generative augmentations are beneficial even in the full dataset regime with many classes. For the final version, we will re run these experiments while also learning augmentation operators, and we expect this version will be even better. We tried 5 datasets: Caltech101, Stanford Cars, FGVC-Aircraft, Stanford Dogs, and Oxford Pets. We also used the following models: Resnet50, Resnet101, VGG19, EfficientNet, Mobilenet, Swin Transformer, Visformer, and Densenet. In almost all cases, the generative models performed better. These results have been included in the appendices, A.6 table 8 (line 1102)
>
>
> For semantic segmentation, we also ran some initial experiments. Our current results show about a 3% improvement on random 5-way, 2-shot subsets of the PASCAL VOC dataset (from 18.77 +- 5.95 percent to 21.53 +- 7.20 percent). We finetuned a Faster R-CNN model with a ResNet-50-FPN backbone from pretrained weights from the COCO dataset. We used one baseline in this case - the PASCAL VOC subset, augmented with classic augmentations. This demonstrates generative augmentations are helpful in this instance as well. For the camera ready paper, we will expand this to more datasets.
>
> We believe that  with these added results that we have sufficiently bolstered our claim.
>
> ---
>
> ## 2. One-shot results and the fitness function inconsistency
>
> > “The claim that ‘EvoAug consistently outperforms the baselines’ contradicts Table 3… This undermines credibility.”
>
> We thank the reviewer for pointing this out. Upon closer analysis, we found that the **Silhouette coefficient and clustering radius contributed unevenly** to the one-shot fitness score (previously \(S - 1/d\)). This imbalance caused suboptimal trees.
>
> To fix this, we added a scaling term which allows us to compute a weighted sum of the Silhouette coefficient and clustering radius. Our old equation was S - 1/d, where S is the Silhouette coefficient and d is the clustering radius. Our new equation is a * S - (1 - a)/d, where a is the variable we use to weight each term.
>
> We then perform a grid search for the weight parameter which balances the two terms, and use the resulting equation as our fitness score.
>
> **This correction significantly improves one-shot performance**, and EvoAug now outperforms the baselines in almost all cases. The updated results are shown in the revised Table 3.
>
> ---
>
> ## 3. Necessity of Generative Operators / Ablation Study
>
> > “No ablation shows that generative operators are necessary… Did the algorithm actually use them?”
>
> We agree this was essential to demonstrate. We added an ablation restricting augmentation trees to Classical + NoOp operators only. EvoAug with full generative operators outperforms this restricted search, confirming:
>
> 1. Generative operators are necessary, and
> 2. The evolutionary search is learning to use them effectively
>    (a random generative tree performs worse).
>
> This ablation is now included in the revised Table 3.
>
> ---
>
> ## 4. Motivation for the One-shot Clustering Metric
>
> > “The justification… is hidden in the appendix, making it appear unmotivated.”
>
> Thank you for the feedback. We  have made the editorial changes you suggested. The motivations should be very clear now.
>
> ---
>
> ## Summary
> In summary, we have:
>
> - **Added extensive new experiments**
>   (five full datasets, eight architectures, segmentation results)
> - **Corrected and improved the one-shot fitness function**, greatly improving accuracy
> - **Added the requested ablation study**, confirming the necessity of generative operators
> - **Made the recommended editorial revisions**
>
> These changes substantially strengthen the empirical and methodological contributions of the paper. We hope that our revisions adequately address your concerns, and we would appreciate if you would consider updating your score.

---

### Official Review · Reviewer_hMoc · 2025-11-02

**Soundness:** 3
**Presentation:** 3
**Contribution:** 3
**Rating:** 6
**Confidence:** 4

**Summary:**

This paper present an automated augmentation strategy that leverages advanced generative models, specifically controlled diffusion and NeRF operators, in combination with classical augmentation techniques. The authors propose a new pipeline called EvoAug for automated data augmentation using generative models and evolutionary search. The key ideas:

1. Traditional augmentation methods (e.g., crop, flip, rotate) are limited in diversity. The authors argue that the recent availability of generative models (e.g., diffusion or few-shot NeRF) allows richer, learned augmentations rather than hand-coded ones.

2. They build an evolutionary algorithm that searches over augmentation trees: hierarchies of stochastic transformations (including generative-model based ones) tailored to a downstream task (e.g., few‐shot classification).

3. Their experiments show that EvoAug can discover augmentations aligned with domain knowledge and yield improved accuracy in fine-grained and few-shot classification tasks under low-data regimes.

A creative, promising direction that bridges generative modeling and automated augmentation. Needs stronger empirical and scalability validation to move from “promising idea” to “established method.”

**Strengths:**

Timely and interesting direction — Combining generative models with augmentation search is appealing, especially for low-data regimes where augmentation matters a lot.

Focus on low-data regime — The fact that they tackle few-shot / fine-grained classification, rather than just standard large‐scale regimes, gives practical relevance.

Domain-alignment — The finding that discovered augmentations “align with domain knowledge” is interesting: suggests the search may rediscover meaningful transformations rather than random noise.

**Weaknesses:**

Limited scalability / generality — The work appears focused on low-data tasks with few-shot classification; it’s unclear how well this would scale to large-scale datasets, high resolution images, or many‐class settings.

Baseline comparisons — It’s unclear if the comparisons include state‐of‐the‐art augmentation methods (e.g., AutoAugment, RandAugment, AugMix) combined with strong generative models, or whether the generative augmentation is compared fairly.

**Questions:**

How is the search space of “augmentation trees” formally defined—are nodes differentiable transformations, or arbitrary generative modules?

---

> ### Author Response · Authors · 2025-12-03
> **Response to Reviewer hMoc**
>
> We thank the reviewer for the constructive feedback and the positive assessment of our contributions. In response:
>
> ---
>
> ## 1. Definition of the Search Space / Differentiability of Nodes
>
> > “How is the search space of ‘augmentation trees’ formally defined—are nodes differentiable transformations, or arbitrary generative modules?”
>
> A key advantage of our approach is that **differentiability is not required**. The nodes in the augmentation tree can be arbitrary generative modules, including non-differentiable ones (e.g., ControlNet, diffusion-based editors, symbolic or procedural operations). Because our optimization is evolutionary rather than gradient-based, the search space flexibly accommodates any operator type.
>
> We have updated the paper to make this more explicit.
>
> ---
>
> ## 2. Baseline Comparisons
>
> > “It’s unclear if the comparisons include state-of-the-art augmentation methods (AutoAugment, RandAugment, AugMix) combined with strong generative models, or whether the generative augmentation is compared fairly.”
>
> To clarify:
>
> - We do compare against strong baselines  with SOTA  AutoAugment and RandAugment baselines with cutmix etc.
> - Our generative approach builds on top of these baselines, so it is certainly both a fair comparison, and a strong baseline.
>
> To further satisfy any concerns, and to help isolate the contribution of augmentation operators, we have added an ablation study to our 5-way, 1-shot experiments. We restrict our augmentation trees to use only classical and NoOp transformation, and run the same EvoAug pipeline. We then record the accuracy of our models trained on the datasets created by these trees. From these experiments, we see that augmentation trees without generative operators tend to score around or worse than the accuracy of random trees. The learned trees with generative operator types outperform those which are restricted to just classical and NoOp operators. This has been included in the results in table 3.
>
> This demonstrates that there is  independent value both in our learnable approach, and  in the generative operators. Leave out either one, and we perfoem worse. Thus, our comparisons are all fair, and our method improves over the baseline no matter how you slice it.
>
> ---
>
> ## 3. Scalability and Generality
>
> > “The work appears focused on low-data settings; it is unclear whether this scales to large datasets, high-resolution images, or many-class settings.”
>
> We included two major sets of new experiments to address this concern.
>
> ---
>
> ### 3.1 Full-Dataset, Many-Class, High-Resolution Classification
>
> We evaluated generative augmentations on five datasets:
>
> - Caltech101
> - Stanford Cars
> - FGVC-Aircraft
> - Stanford Dogs
> - Oxford-IIIT Pets
>
> Across eight architectures, including ResNet-50/101, VGG19, EfficientNet, MobileNet, Swin Transformer, Visformer, and DenseNet, generative augmentations improved accuracy in nearly all settings.
>
> For the rebuttal period, we used **ControlNet-only operators due to compute constraints of learning augmentation trees for larger datasets, just to demonstrate the basic method. For the camera-ready version, we will run full augmentation-tree search, which we expect to perform even better. These results now appear in **Appendix A.6, Table 8 (line 1102)**.
>
> ---
>
> ### 3.2 Semantic Segmentation
>
> We also conducted segmentation experiments on **PASCAL VOC** (5-way, 2-shot subsets):
>
> - Baseline (classic augmentations): **18.77 ± 5.95%**
> - Generative augmentations (ours): **21.53 ± 7.20%**
>
> We fine-tuned a Faster R-CNN with a ResNet-50-FPN backbone pretrained on COCO. This demonstrates that generative augmentation also benefits **dense prediction tasks**. For the camera ready paper, we will expand this to more datasets.
>
>
> ---
>
> ## 4. Final Note
>
> We thank the reviewer again for the valuable feedback. We hope that the added ablation study, the full-dataset experiments across diverse architectures, and the new segmentation results adequately address your concerns regarding fairness, generality, and scalability.
>
> If these clarifications and additional experiments resolve your reservations, we would greatly appreciate if you would raise your score.

---

### Author Response · Authors · 2025-12-03
**Final Remarks for Discussion Period**

## Reviewers:
Thank you for your thoughtful, detailed, and constructive feedback. Your comments substantially improved the clarity, rigor, and empirical strength of our manuscript. We wish that we would have been able to continue the discussion the intended way. Nevertheless, we are grateful for the discussions we were able to have and the improvements we have made with your help. Our work is undoubtedly strengthened by your input.



## Area Chairs:
We hope you find our proposed EvoAug framework to be a meaningful contribution to automated data augmentation and the integration of generative models into learning pipelines.


During the rebuttal period, we responded comprehensively to every comment and concern. We added entirely new sets of experiments—including full-dataset, many-class evaluations across eight architectures, and semantic segmentation results, showing that generative augmentations provide consistent improvements in these settings. We performed the requested ablation isolating the effect of generative operators, demonstrating that both generative augmentations, and the augmentation tree learning method, are necessary for the gains observed. We also fixed the imbalance in our one-shot fitness metric, which substantially improved results and resolved earlier inconsistencies. Finally, we clarified terminology, added missing literature, and revised text where misunderstandings had arisen. We believe these expanded results and methodological refinements address the reviewers’ core concerns.

---

### Meta-Review · Area_Chair_ty5Y · 2026-01-06

**Summary:**

This paper proposes EvoAug, a data augmentation approach using generative models. They define an evolution strategy where an augmentation is defined using a binary tree of operations as nodes. For operations, they use diffusion-based operators (ControlNet) and condition with Canny edge detection, Segment Anything segmentations, and depth maps from MiDaS as well as NeRF-based augmentation (Zero123). For fitness functions, generally, an augmentation tree can be evaluated by simply training a model with generated augmentations on the training data and measuring performance on the previously unseen evaluation data. They consider two special settings: low-data setting and one-shot setting. In low-data settings they tackle the noise of evaluation using K-fold cross-validation and measuring the negative validation loss as the fitness function. In one-shot setting, they consider three strategies: Label-Efficient Clustering,
Double Augmentation to increase data, and measuring Training Loss. Their experiments compare their strategy to three baselines: random application of classical augmentations, RandAugment with a grid search, and AutoAugment.

Reviewers praised the creative use of conditional generative models for automated augmentation search, the introduction of clustering-based and loss-based fitness functions. The rigorous approach and focus on the low-data regime were also highlighted as strengths.

**Reviewer Concerns:**

Reviewer UETn
- **Modest and dataset-dependent performance improvements.** The reviewer notes that the results underperform standard augmentations for some datasets. The authors provide additional results in Appendix A.6, Table 8 (line 1102) where they observe improved accuracy in two settings: full-dataset, many-class, high-resolution classification, and semantic segmentation. The authors argue while their method is not always better, it more often than not improves over strong baselines.
- **Using diffusion and NeRF models for each augmentation is expensive**: The authors acknowledge this limitation but note that it is a one-time upfront cost and the 24h duration is because of their old hardware. The AC notes that this answer does not address the concern about inference overhead of using these augmentations and a measurement of the wall-clock time overhead is required.
- **Learning augmentations for full-scale datasets is prohibitively slow**: The authors agree but argue it is still possible to learn using a subset and refer to Appendix A for a demonstration.
- **Terminology of unsupervised clustering metric**: The authors have revised the paper to refer to it as label-efficient clustering.
- **Missing references**: The authors incorporated references into the paper.
- **Missing ablation for the usefulness of generative operators**: The authors provide an ablation by restricting augmentation trees to Classical+NoOp operators only included in Table 3.
- **Missing ablation for trees deeper than depth=2**: The authors started running ablations but could not provide results for the rebuttal.
- **Comparison of one-shot fitness variants**: The authors found an imbalance in their one-shot fitness score and provided improved results.
- **Add figures illustrating example augmented images**: The authors did not respond.
- **Comparison to other modern augmentation/search techniques (e.g. ASHA, PBA)**: The authors did not have time to run this during the rebuttal.

Reviewer BU6y:
- **Missing sensitivity analysis on population size, mutation rate, depth-2 trees**: The authors did not have enough time to run these during the rebuttal.
- **Contribution of each augmentation operator type (diffusion, NeRF, classical) is not isolated**: The authors point to new ablation results restricting augmentation trees to Classical + NoOp.
- **Adaptability to other tasks (e.g., detection, segmentation)**: The authors provide additional results for two settings: Full-Dataset, Many-Class, High-Resolution Classification, and Semantic Segmentation.
- **Cross-domain transferability**: The authors defer this study to future work.

Reveiwer VpwN:
- **Mixed Improvements in 2-shot and 5-shot Results.** The reviewer notes that the results underperform standard augmentations for some datasets in Tables 1-2. The authors provide additional results in Appendix A.6, Table 8 (line 1102) where they observe improved accuracy in two settings: full-dataset, many-class, high-resolution classification, and semantic segmentation.
- **Results in one-shot setting are not positive**: The authors found an imbalance in their one-shot fitness score and the correction improves one-shot performance.
- **Missing ablations for the benefit of multiple expensive generative operators (Canny, Color, Depth, Segment, NeRF)**: The authors added an ablation restricting augmentation trees to Classical + NoOp operators only.
- **Justification for the 1-shot clustering fitness function**: The authors made an edit to the paper to address.

Reviewer hMoc:
- **Limited scalability / generality**: The authors included new results.
- **Baseline comparisons**: The authors note that they already compare with AutoAugment and RandAugment while referring to their new ablation results.
- **Are nodes differentiable**: The authors note the nodes do not need to be differentiable.

**Reviewer Scores:**

The reviewers gave scores of 2, 4, 6, 6. The reviewers generally found the approach creative. The major concerns can be summarized as mixed performance gains in some settings depending on the dataset and limited ablations. The authors provided new ablations and improvements to their method during the rebuttal. However, the AC finds that the results are still dataset dependent with no clarity on when to use the proposed method on a particular dataset. This is particularly concerning as the paper has not demonstrated cross-domain transferability. Moreover, the overhead of the method during the training has not been analyzed. The authors discuss the overhead of their method as a one-time cost, however, the final augmentation is still more complex than standard augmentations and its overhead needs to be measured and analyzed. Other unaddressed limitations include missing ablations such as the effectiveness of individual operations. The paper would benefit from a more detailed analysis of the final augmentation trees found for each dataset and visual examples of augmented inputs. Additionally, usefulness of the method for larger datasets and other tasks would improve the work.

---

### Decision · Program_Chairs · 2026-01-26

Reject